# Digital transformation and total factor productivity: Empirical evidence from China

Zhonghao Lei[1], Dongmei Wang[2]*

1 School of Economics, Lanzhou University, Lanzhou, Gansu, China, 2 Research Center for Economy of Upper Reaches of the Yangtze River, Chongqing Technology and Business University, Chongqing, China

* wdm_gs@126.com

**Data Availability Statement:** All minimal anonymized data set necessary to replicate our study findings files are available from the Qualitative Data Repository database (accession number(s) https://doi.org/10.5064/F6QPPKC4.

## Abstract

Digital transformation plays a crucial role in improving the quality development of companies in this era of digital economy with ever-changing technologies. This paper empirically investigates the impact of corporate digital transformation on total factor productivity and the mechanism of action, using A-share listed companies in Shanghai and Shenzhen from 2011–2021 as the research sample, and found that the digital transformation of companies significantly improves total factor productivity, with the plausibility of the findings being verified by a series of robustness tests. Based on the heterogeneity study, it is found that such effect is stronger for private companies, non-high-tech companies, and companies with a high degree of industry competition. The mechanism test indicates that digital transformation facilitates total factor productivity through four ways: strengthening company technological innovation, reducing operational costs, increasing resource allocation efficiency, and improving human capital structure. The findings of this paper support a better understanding of the micro effects of digital transformation and provide empirical evidence for policy formulation and adjustment.

## 1. Introduction

The reform and opening up policy in China started over four decades ago has allowed a long-term rapid economic growth in the country with inputs on capital, labor, land and other factors. However, such outstanding issues as a *large yet not strong* manufacturing industry, low technological innovation level and uncompetitive practices on international market of companies have restricted its quality economic development. With the changes of demographic structure, labor cost and supply and demand, the development method of relying solely on factor inputs to promote economic growth became unsustainable. Total factor productivity is an important indicator to evaluate the quality of economic development and is the main source of sustainable economic development [1], and enhancing total factor productivity is an important way to promote the growth of economic aggregate and achieve high quality economic development.

In recent years, the Chinese government has attached great importance to the development of digital economy and promulgated a series of relevant policies to promote the in-depth

**Funding:** The 2023 Gansu Provincial Department of Education College Teachers' Innovation Fund Project (No. 2023A-281) from China and Research Platform Open Subject of Chongqing Technology and Business University (grant number: KFJJ2022039) and the Chongqing Graduate Student Research and Innovation Program 2023 (grant number: CYB23263). The funder was involved in the study design as well as the final finalization and publication decisions.

**Competing interests:** The authors have declared that no competing interests exist.

integration of digital economy and real economy. Correspondingly, China is moving into the digital era as its digital governance system is gradually established and digital resources have become an important production factor of economic activities, with the rapid development of emerging digital technologies such as big data analysis, artificial intelligence, Internet of Things, cloud computing and blockchain, as well as the continuous expansion of related application scenarios and the continuous integration and innovation of cutting-edge digital information technology and industrial mechanism models. According to the China Digital Economy Development Report (2022), China's digital economy would reach 45.5 trillion yuan in 2021, with a nominal growth of 16.2% year-on-year, highlighting the role of the digital economy as a "gas pedal" and "stabilizer" of the national economy. As a result, we considered two issues in this paper—will digital transformation be able to improve the total factor productivity level of companies and what the impact mechanisms are, and the research supports proper assessment on the economic effect of digital transformation at the micro level. An in-depth investigation of the impact of digital transformation on total factor productivity can effectively help policy makers to implement policies precisely, and expand the research on the micro impact related to the digital economy.

Current literature examining the economic impact of digital transformation demonstrates valuable findings. The first is the definition of the concept of digital transformation [2]. In generally, companies start to introduce such emerging digital technologies as cloud computing, artificial intelligence, Internet of Things, and big data into their production and operation to adapt to digital economy as they positively improve the production models, management processes, business behaviors, and internal organizational culture, which raises their market competitiveness and accelerates their transformation and upgrading. This series of transformation process is digital transformation [3]. Digital transformation is not simply the use of digital technology in business, but the reshaping of business processes and production models by digital technology, which has a profound impact on business performance, management systems, human capital structure, and investment efficiency [4–7]. Some scholars point out that digital transformation represents a new model for the development of company informatization, which is a result of the profound integration of company production and operation and new digital technologies [8]. Notably, digital transformation brings all-round changes in production, business operations, and internal organizational management of companies [9]. Some existing literature focuses on the microeconomic effects of digital transformation, with some suggesting that digital transformation of companies can effectively improve business performance [10]. The large-scale investment of potential general-purpose technologies, such as artificial intelligence, into business operations can stimulate companies to achieve technological progress [11–13], improve productivity [14], and thus raise company value and capital market performance [15]. At the company management level, digital transformation can improve corporate governance, operational efficiency, and information processing capabilities [16], as well as facilitating the collaboration efforts among different companies and accelerating the rate of innovation in business models and business structures [17]. Some literature focuses on the impact of digital transformation of companies on productivity, with some scholars arguing that digital transformation of companies is an exogenous shock under established resource constraints and that extensive use of digital technologies by companies can reduce production costs and increase productivity [18, 19].

A large number of valuable studies has been conducted on the development of digital transformation of companies and the economic impact generated by digital transformation. This paper attempts to explore the microeconomic impacts arising from digital transformation from a productivity perspective. Compared with the existing related literature, the marginal contribution of this paper may be reflected in the following two aspects: first, this paper

analyzes the impact of digital transformation on total factor productivity of companies from a multidimensional perspective. On the one hand, this paper investigates the overall impact of digital transformation on total factor productivity and conducts robustness tests on it, while we construct instrumental variables and apply two-stage least squares to deal with possible endogeneity issues. On the other hand, this paper analyzes the channels of digital transformation on total factor productivity from four aspects: technological innovation, operation cost, resource allocation efficiency, and human capital. Second, this paper explores the heterogeneous effects of digital transformation on total factor productivity from the property rights nature of companies, the degree of technological innovation, and the degree of industry competition to provide more precise theoretical basis for government policy makers.

## 2. Theoretical mechanism analysis

First, digital transformation improves the total factor productivity of companies by promoting technological innovation. Technological innovation is an important influencing factor of total factor productivity of companies [20]. Digital transformation changes the innovation model and innovation process of companies, and increases the innovation efficiency and R&D scale. Analysis at the innovation model level indicates that digital transformation prompts individual companies and departments to shift from the traditional closed innovation development model to an open and networked collaborative innovation model [21]. Companies have strengthened their information development using emerging digital information technologies such as big data, cloud computing, artificial intelligence, and the Internet of Things, which have significantly improved the efficiency and effectiveness of communication within and between companies. The interconnection of key data and technologies accelerates the formation of knowledge spillover effects, which in turn promotes collaborative innovation in various departments within the company. In addition, industrial Internet serves as a new infrastructure, application model and industrial ecology for the deep integration of new-generation information and communication technology and industrial economy. Companies build up a new manufacturing and service system covering the whole industrial chain and the whole value chain through the industrial Internet platform, integrate the production subjects of each link of the supply chain network with high efficiency, establish a mechanism for sharing data resources in the industry chain and supply chain and realize cross-company, cross-industry and cross-company, cross-industry, and cross-region networked collaborative innovation R&D new model to improve the overall innovation efficiency [22]. Analyzing from the level of innovation process, as companies and industries gradually improve the level of digital transformation, the problems of backward industrial design software and low R&D efficiency of manufacturing companies are resolved. Companies improve the process of product design and R&D through digital technologies and visualization tools, which in turn improve the output efficiency of R&D inputs. The application of AIGC (artificial intelligence generated content) in industries like media is a typical example, with the continuous development and progress of data, computing power, and algorithms, generative AI empowers production companies to develop underlying technical models and reduce the workload of manual modeling [23]. Based on this, it is easier for companies to achieve fine-grained and modular control over the operational processes of new product development and emerging technology development.

Second, digital transformation enhances total factor productivity by reducing operational costs. On the one hand, the deep integration of digital technologies into production and operation processes significantly increases the efficiency of production tools, and intelligent devices collect specific data according to program commands, and then analyze and make decisions

on these data, which significantly reduces manual execution costs and costs of operation and maintenance, and ultimately improves productivity [24]. For example, companies integrate data related to operation business, management system and supply chain based on intelligent infrastructure such as industrial Internet platform and Internet of Things, promote sharing of key data, while comprehensively collecting and analyzing information from all aspects of production, purchase and sales of the company to achieve refined management of the whole product life cycle, thus reducing operational costs and improving management efficiency. On the other hand, the increasing maturity of digital information technology for collection, transmission, interaction and storage renders the information structure within the company to become instant, detailed, continuous and complete. Such information promotes companies to gradually establish a modular and open company management model and a shared network platform in the process of digital transformation [25], which allows companies to realize highly collaborative real-time information interaction both internally and externally, increase interdepartmental synergy, strengthen the management level of managers, and thus reduce the cost of information exchange.

Third, digital transformation improves resource allocation efficiency and thus enhances total factor productivity of companies. This paper analyzes the role of resource allocation efficiency in the process of digital transformation affecting company total factor productivity from the perspective of labor investment efficiency and supply chain resource operation efficiency. From the former perspective, labor is an indispensable input factor in the production and operation process of companies, and its allocation in companies profoundly affects their sustainability and operational efficiency [26]. The in-depth application of emerging digital technologies within companies provides an optimized channel for information transfer and resource integration between departments [27]. Digital transformation facilitates information transfer, information exchange, and information integration among departments, and enhances the information transparency of companies. With greater information transparency in a company, the supervision from the shareholder and regulatory departments becomes stronger and the constraints on management decisions are greater; therefore, the management level are refrained from making such labor investment decisions as over- or under-employment caused by excessive focus on short-term interests [26]. Thus, digital transformation improves information transparency within the company, which facilitates management to make optimal labor employment decisions and discourages inefficient investments, supporting companies to improve total factor productivity. We then analyze the mechanism of resource allocation efficiency from the perspective of supply chain resources. Increasing application of digital technology to the fundamental processes of the supply chain allows the digital transformation of companies to significantly lower information asymmetry among various departments of companies and between companies and other suppliers [28]. Digital technology accelerates the information exchange between upstream and downstream companies in the supply chain, and the information feedback mechanism based on digital technology enables the information interconnection between companies, realizing the precise matching of supply and demand and an improved efficiency of matching supply and demand in the supply chain. Additionally, digital transformation supports companies to reshape their supply chain inventory management mode. Manufacturers can use emerging digital technologies such as blockchain, cloud computing, and big data to accurately, to predict changes in market demand [29], timely adjust inventory levels, and reduce excessive stagnant inventory and stale materials. Furthermore, companies implementing digital transformation initiatives show improvement in overall information transparency of the supply chain. The digital technology featured by blockchain can be easily applied to supply chain operation scenarios, where authorized companies are allowed to supervise the whole process from product production to market sales

through such blockchain technologies as distributed ledger, asymmetric encryption, consensus mechanism, and smart contracts, which substantially reduced the risk of default of upstream and downstream companies in the supply chain, and supports companies to arrange production, inventory, and sales plans more rationally.

Fourth, digital transformation affects total factor productivity by improving human capital structure of companies. While it creates new productivity, production and business models, digital transformation also "disrupts" the old ones [30]. As new infrastructures are built across the entire industry and value chain, new application models and new digital technologies continue being used, some conventional job positions and work sectors will be replaced by intelligent industrial equipment and digital platforms, and the demand for low-end labor will decrease while the demand for high-end labor, increase [31]. The change in labor demand leads to the upgrading of the human capital structure of companies, which is reflected in the increase of the proportion of high-skilled labor in companies. After upgrading the human capital structure of companies, the high quality human capital and skill knowledge reserves will adjust the original production and operation mode and enhance the knowledge spillover effect, thus improving the efficiency of company production and operation.

## 3. Model design and variable description

### 3.1 Benchmark model design

In this paper, we explore whether the digital transformation of companies affects total factor productivity. Based on the individual and temporal heterogeneity of companies, we refer to other literature [4] to construct the following benchmark regression model:

$$TFP_{it} = \beta_0 + \beta_1 DT_{it} + \gamma X_{it} + \lambda_t + \eta_i + \varepsilon_{it} \tag{1}$$

where the subscripts $i$ and $t$ denote company and year, respectively. The dependent variable $TFP_{it}$ represents the total factor productivity of the company. $DT_{it}$ is the digital transformation of the company, and the parameter $\beta_1$ measures the impact of digital transformation on total factor productivity. $X_{it}$ represents a series of company-level control variables, and $\gamma$ is the impact coefficient corresponding to the control variables. $\lambda_t$ represents year fixed effects, and $\eta_i$ represents individual fixed effects. To ensure that the regression results are more robust, we adopt a cluster robust standard error regression model for estimation.

### 3.2 Variable definition

**3.2.1 Explained variables (TFP).** Empirical studies in the micro domain usually use the semiparametric treatment proposed by Olley and Pakes (1996) [32] as well as the semiparametric treatment proposed by Levinsohn and Petrin (2003) [33] to measure the total factor productivity of firms, where the OP method requires that company investment and productivity shocks to satisfy a strong monotonic relationship. Discarding the sample with zero current investment would result in missing sample data and sample selection problems. In this paper, we adopt the LP method to measure the total factor productivity level of companies, as it uses the intermediate goods input index instead of company investment amount as a proxy variable. The annual operating revenue represents the total output of the company, the net fixed assets represent the capital input, the number of employees of the company represents the labor input variable, and the intermediate input is represented by adding the operating cost of the company, the selling expense, administrative expense, and financial expense of the company and deducting the current depreciation and amortization of the company and the cash paid to and for the employees.

**3.2.2 Core explanatory variables (DT).**  The existing literature generally takes two approaches to measure the degree of digital transformation of companies: one is to measure the degree of digital transformation of companies by the inputs related to digital transformation [2], and the other is to measure the degree of digital transformation by textual indicators, using the frequency of keywords related to digital transformation in annual reports [34]. What is reflected through keywords in the annual report are the perceptions and predictions of industry trends given by the senior management team of the company, which may not accurately depict the degree of digital transformation of the company at present. In this paper, we choose to use the first method in the benchmark regression to construct corporate digital transformation indicators. The specific approach is as follows: when the year-end intangible assets disclosed in the annual financial report of each listed company include keywords related to digital technology and patents, such as management system, intelligent platform, network, software, client, we define this item as the digital intangible assets of the company, which are summed and then divided by the amount of intangible assets of the company to obtain the proxy variable for the digital transformation of the company.

The control variables include company growth capacity *Growth*, solvency *Cashflow*, profitability *Roa*, gearing ratio *Lev*, company size *Size*, company age *Age*, majority shareholder ownership *Tpo*, management ownership *Mshare*, Tobin's Q *TobinQ*, and independent director ratio *Indep*. The specific definitions of the variables are shown in Table 1.

### 3.3 Data description

This paper selects Chinese listed companies in Shanghai and Shenzhen A-shares from 2011–2021 as the research sample, and the data sources are the annual reports of companies from the websites of Shanghai Stock Exchange and Shenzhen Stock Exchange, used to calculate the digital transformation of companies in this paper, and Wind database and CSMAR database

**Table 1. Variable descriptions.**

| Variable Symbols | Variable Name | Variable construction methods |
|---|---|---|
| *TFP* | Total Factor Productivity | LP method measurement |
| *DT* | Digital Transformation | Intangible assets related to digital technology as a percentage of overall corporate intangible assets |
| *Growth* | Business Growth Capability | Expressed in terms of operating income growth rate, (Operating income for the year, Operating income of the previous year)-1 |
| *Cashflow* | Solvency | Expressed as cash flow ratio, net cash flow from operating activities divided by total assets |
| *Roa* | Profitability | Expressed as return on total assets, net income divided by total assets |
| *Lev* | Gearing ratio | Total liabilities at the end of the year divided by total assets at the end of the year |
| *Size* | Enterprise size | Natural logarithm of total assets at the end of the year |
| *Age* | Business Age | Ln (Current Year—Year of company establishment +1) |
| *Tpo* | Shareholding ratio of major shareholders | Total number of shares held by the top five largest shareholders divided by the total number of shares |
| *Mshare* | Management shareholding ratio | Number of shares held by management divided by the total number of shares |
| *TobinQ* | TobinQ | (Market value of shares outstanding + Number of non-marketable shares×Net assets per share + Carrying value of liabilities)/Total Assets |
| *Indep* | Percentage of independent directors | Number of independent directors divided by the number of directors |

**Table 2. Descriptive statistics.**

| Variables | Sample size | Average value | Standard deviation | Minimum value | 25th percentile | Median | 75th percentile | Maximum value |
|---|---|---|---|---|---|---|---|---|
| TFP | 28438 | 9.09 | 1.12 | 5.62 | 0.00 | 8.99 | 9.75 | 13.64 |
| DT | 28438 | 0.10 | 0.22 | 0.00 | 0.00 | 0.01 | 0.06 | 1.00 |
| Growth | 28432 | 0.17 | 0.43 | -0.66 | -0.02 | 0.11 | 0.27 | 4.33 |
| Cashflow | 28438 | 0.05 | 0.07 | -0.20 | 0.01 | 0.05 | 0.09 | 0.26 |
| Roa | 28437 | 0.04 | 0.07 | -0.40 | 0.01 | 0.04 | 0.07 | 0.26 |
| Lev | 28438 | 0.43 | 0.21 | 0.03 | 0.27 | 0.42 | 0.58 | 0.93 |
| Size | 28438 | 22.22 | 1.28 | 19.50 | 21.32 | 22.04 | 22.94 | 26.45 |
| Age | 28438 | 2.90 | 0.33 | 1.39 | 2.71 | 2.94 | 3.14 | 3.61 |
| Top | 28438 | 0.53 | 0.15 | 0.18 | 0.42 | 0.53 | 0.64 | 0.89 |
| Mshare | 27718 | 0.14 | 0.20 | 0.00 | 0.00 | 0.01 | 0.25 | 0.71 |
| TobinQ | 27909 | 2.07 | 1.42 | 0.80 | 1.25 | 1.63 | 2.35 | 17.73 |
| Indep | 28436 | 0.38 | 0.05 | 0.29 | 0.33 | 0.36 | 0.43 | 0.60 |

for all data on company-level indicators. The reliability of the empirical findings is assured as we removed the sample data of the financial and real estate industries, companies with abnormal data and substantial data missing, and the sample data of ST or *ST and companies that issue both A and B shares. Meanwhile, to prevent extreme outliers from interfering with the regression results, we performed tailoring at the 1% level for all continuous variables of company characteristics in this paper. Table 2 shows the descriptive statistics of the variables in the benchmark regression.

## 4. Empirical analysis

### 4.1 Benchmark regression analysis

Table 3 shows the estimation results of the benchmark regression model (1). In column (1), we include company-level control variables, and the estimated coefficient of the explanatory variable $\beta_1$ is 0.2938, which is significantly positive at the 1% level. In column (2), we control individual fixed effects in column (1) and the estimated coefficient of the explanatory variable $\beta_1$ is also significantly positive at the 1% level. In column (3), we control both individual fixed effects and time fixed effects, and the estimated coefficients of the explanatory variables $\beta_1$ are also significantly positive at the 1% level, which reflects the regression results of the benchmark regression model (1). We take the results in column (3) as an example, and on average, each

**Table 3. Benchmark regression.**

| Explanatory Variable | Explained variable | | |
|---|---|---|---|
| | **(1)** | **(2)** | **(3)** |
| DT | 0.2938*** (0.0201) | 0.1013*** (0.0328) | 0.0919*** (0.0332) |
| Constant term | -5.6839*** (0.0873) | -4.4863*** (0.2801) | -5.1418*** (0.3735) |
| Control variables | YES | YES | YES |
| Individual fixed effects | NO | YES | YES |
| Year fixed effects | NO | NO | YES |
| N | 27188 | 26863 | 26863 |
| $R^2$ | 0.7072 | 0.9365 | 0.9387 |

Notes: The robustness standard errors

*, **, *** in brackets represent the significance levels of 10%, 5% and 1%, respectively. These are consistent in the following tables.

unit increase in the degree of digital transformation of companies is associated with a 9.19% increase in total factor productivity. The estimated results can statistically indicate that digital transformation significantly raises the level of total factor productivity of companies.

## 4.2 Endogenous handling

**4.2.1 Instrumental variables method.** In general, companies with higher total factor productivity have stronger financing and technology development capabilities, and greater ability and willingness to achieve digital transformation. Potentially omitted variables and endogeneity issues due to two-way causality can lead to bias in the estimated coefficients of the benchmark regression. To tackle the endogeneity problem, we re-estimate the benchmark regression model using the instrumental variables approach. Referring to other literature [35], this paper uses the mean value of digital transformation of listed companies located in the city and the mean value of digital transformation degree of listed companies in the industry as the instrumental variables of the company's digital transformation variables for two-stage least squares (2SLS) estimation, respectively. We choose the mean of digital values at the city and industry level as IVs for two reasons: first, the degree of digital transformation of individual firms is affected by the digital transformation of the region and industry in which the firms are located, so the instrumental variables we choose satisfy the correlation assumption. Second, the regional mean and industry mean of firms' digital transformation have been homogenized and are weakly correlated with firms' total factor productivity. In this sense, the instrumental variables we selected are not related to the current production and business activities of the firms and satisfy the exclusivity assumption. The estimation results are shown in columns (1) and (2) of Table 4. In column (1), the instrumental variable used in the estimation model is the mean value of digital transformation of listed companies in the city where the company belongs, and the estimated coefficient of the explanatory variable $\beta_1$ is significantly positive at the 1% level. In column (2), the estimated model selects the mean value of digital transformation of listed companies in the company's industry as the instrumental variable, and similarly, the estimated coefficient of the explanatory variable $\beta_1$ is also significantly positive at the 1% level. We next tested the plausibility of the instrumental variables. The first is to test the correlation hypothesis, the first stage regression results estimated by 2SLS show that the coefficients of the regional mean and industry mean of enterprise digital transformation are 0.9792 and 0.6283, respectively, which both pass the significance test at the 1% level, indicating that the instrumental variables are strongly correlated with the core explanatory variables. We then test whether the exclusivity assumption is valid. In general, the exclusionary hypothesis is difficult to test directly, and the presence of a weak instrumental variable problem will invalidate the exclusionary hypothesis. Therefore, we focus on the 2SLS estimated correlation statistic to

**Table 4. Endogenous treatment.**

| Explanatory Variables | Explained Variables | | | |
|---|---|---|---|---|
| | **(1)** | **(2)** | **(3)** | **(4)** |
| | **IV** | **IV** | **Lagged first-order regression** | **Lagged second-order regression** |
| *DT* | 0.8623*** (0.1730) | 0.4836*** (0.1437) | 0.0479* (0.0247) | 0.0372 (0.0355) |
| Control variables | YES | YES | YES | YES |
| Individual fixed effects | YES | YES | YES | YES |
| Year fixed effects | YES | YES | YES | YES |
| *N* | 27188 | 27188 | 23066 | 19772 |
| $R^2$ | 0.7467 | 0.7560 | 0.9379 | 0.9355 |

determine whether there is a weak instrumental variable problem. The Cragg-Donald Wald F-statistic and Kleibergen-Paap Wald rk F-statistic of the above two instrumental variables are both greater than the critical value of 10% bias in the Stock-Yogo weak ID test critical values, and the original hypothesis can be rejected, and it can be considered that there is no weak instrumental variable problem and the paper selected appropriate instrumental variables. It should be emphasized that the IVs estimates obtained in this paper through instrumental variables are local average treatment effects that exclude the part of exogenous changes included in the explanatory variable digital transformation. That is, changes in firms' digital transformation due to bidirectional causation are excluded, and thus the regression coefficients are significantly different from the OLS coefficient values. In addition, because the endogeneity issue does affect the study estimates, we use IVs estimation in both the heterogeneity test and the mechanism test later.

**4.2.2 Explanatory variables lagged.** Considering that there is a lag in the impact of digital transformation on company production and operation, the current period values of the explained variables generally do not affect the lagged terms of the explanatory variables, and no two-way causality exists. Therefore, in this paper, the core explanatory variables *DT* in the benchmark regression model are treated with one and two lags before being regressed separately, and the estimated results are shown in columns (3) and (4) of Table 4. When the core explanatory variable is the first-order lagged term of company digital transformation, the estimated coefficient is significantly positive at the 10% level, indicating that there is a lag in the effect of digital transformation initiatives on total factor productivity improvement. When the core explanatory variable is the second-order lagged term of digital transformation of companies, the estimated coefficient is insignificantly positive, which indicates that digital transformation initiatives have a poor effect on the total factor productivity of companies two years into the future.

## 4.3 Robustness test

**4.3.1 Substitution of explained variables.** In response to the potential covariance of total factor productivity estimated by the OP and LP methods, the ACF method improves the first stage estimation of the OP and LP methods [36]. The paper uses the ACF method to re-measure the total factor productivity of listed companies and regresses it as a proxy for the explained variables, the other control variables are consistent with the benchmark regression, and the estimated results are shown in column (1) of Table 5. The estimated coefficient of digital transformation is significantly positive at the 10% level, which is basically the same as the estimated result of the benchmark regression.

**4.3.2 Substitution of core explanatory variables.** In the robustness check section, we use the frequency of words related to "digital transformation" in the annual reports of listed

**Table 5. Robustness tests.**

| Explanatory Variables | Explained variable | | | | | |
|---|---|---|---|---|---|---|
| | **(1)** | **(2)** | **(3)** | **(4)** | **(5)** | **(6)** |
| *DT* | 0.0585* (0.0330) | 0.0273*** (0.0051) | 0.0643** (0.0320) | 0.0955** (0.0475) | 0.0967** (0.0400) | 0.0920* (0.0427) |
| Constant term | -2.8760*** (0.3203) | -5.2582*** (0.3721) | -5.1348*** (0.3520) | -2.3410*** (0.3442) | -5.2320*** (0.4028) | -5.1418*** (0.5099) |
| Control variables | YES | YES | YES | YES | YES | YES |
| Individual fixed effects | YES | YES | YES | YES | YES | YES |
| Year fixed effects | YES | YES | YES | YES | YES | YES |
| Industry fixed effects | NO | NO | YES | NO | NO | NO |
| *N* | 27581 | 25732 | 27188 | 17778 | 22352 | 26863 |
| $R^2$ | 0.9210 | 0.9320 | 0.7168 | 0.9604 | 0.9364 | 0.9382 |

companies to measure the degree of digital transformation. We extract all the text contents from annual reports of all listed companies in China's Shanghai and Shenzhen A-share markets, then we search, match, and count the frequency of the terms in the text related to the five aspects of artificial intelligence technology, big data technology, cloud computing technology, blockchain technology, and digital technology application, and sum up the frequency of the five aspects as the digital transformation index. As the word frequency data has the characteristic of "right bias", we use the summed data as a proxy variable for digital transformation after natural logarithm processing. The regression results are shown in column (2) of Table 5, and the estimated coefficients of the explanatory variables are significantly positive at the 1% level, which is consistent with the results of the benchmark regression.

**4.3.3 Adding control variables.** In the robustness test, we control industry fixed effects while controlling individual fixed effects and year fixed effects, and the other control variables remain consistent with the benchmark regression model. The specific estimation results are presented in column (3) of Table 5, and the estimated coefficient of digital transformation is significantly positive at the 5% level.

**4.3.4 Subsample regression.** First, this paper conducts subsample regressions for industry classification. The manufacturing industry is an integral part of the national economy, and the Chinese government released the strategic plan "Made in China 2025" in 2015, with an aim to promote the transformation and upgrading of its manufacturing industry and improve the intelligent degree of manufacturing companies. Compared with the service, mining, and agriculture industries, digital transformation has a greater impact on the production and operation of the manufacturing industry. Here we conduct a separate subsample regression for manufacturing companies, and the estimated results are reported in column (4) of Table 5. The estimated coefficient of digital transformation is significantly positive at the 5% level, indicating that the implementation of digital transformation measures in manufacturing companies can significantly improve total factor productivity.

Second, a municipality directly supervised by the Central Government has a large built-up area and a large resident population, and is ahead of other prefecture-level cities in terms of political status, level of economic development, industrial production capacity, innovation activity, level of openness, and infrastructure development. Companies located in municipalities have more resources and access to information, which can be used to improve their productivity levels in other ways. We remove the sample of companies located in municipalities and regress the remaining sample of companies, and the empirical results are shown in column (5) of Table 5. The estimated coefficient of digital transformation is significantly positive at the 5% level, indicating that the benchmark regression results are robust.

**4.3.5 Double clustering robust standard error.** According to relevant literature, when the regression variables contain important information about time and individual characteristics, it is more efficient to adopt the robust standard error of double clustering of company and year than single clustering [37], and here we replace the single clustering estimation of the benchmark regression with the double clustering estimation of company-year, and the regression results shown in column (6) of Table 5. It can be referred that even after adjusting the clustering levels, the estimated coefficients of the explanatory variable digital transformation are still significantly positive, and the magnitude of the coefficients is largely consistent with the benchmark regression, demonstrating the robustness of the empirical results.

## 4.4 Heterogeneity test

**4.4.1 Corporate heterogeneity test.** The estimation results of the benchmark regression and subsequent robustness tests suggest that digital transformation initiatives can significantly

improve the total factor productivity of companies, but such influence may have heterogeneous effects due to differences in the underlying characteristics of companies. It is worth noting that in order to mitigate the impact of endogeneity issues on the results as much as possible, we used the 2SLS model for the regression in the heterogeneity test, and the instrumental variable was the regional mean of the firm's digital transformation.

First, this paper examines the heterogeneous impact of digital transformation based on the differences in the nature of ownership of companies, and regressions are conducted for state-owned companies and private companies in groups. The specific regression results are shown in columns (1) and (2) of Table 6, where column (1) presents the regression results for the subsample of state-owned companies, and the estimated coefficient of digital transformation is not significantly positive, while column (2) presents the regression results for the subsample of private companies, and the estimated coefficient of digital transformation is significantly positive at the 1% level, and the value is larger than the estimated coefficient of the sample of state-owned companies, indicating that compared with state-owned companies, the effect of digital transformation on total factor productivity is stronger for private companies. We explain this phenomenon from the perspective of organizational inertia theory. Digital transformation is a top-down internal reform in companies, and a stronger organizational inertia is formed within SOEs compared with private companies, which generates resource inertia and behavioral process inertia that inhibit companies from accessing new knowledge and using new production and operation models, reducing the efficiency of companies. The final result is that the organizational inertia of SOEs hinders the positive effect of digital transformation on productivity. Private companies have a higher degree of flexibility in their management and production systems and encounter less resistance to implement digital transformation measures.

Second, this paper tests the heterogeneous impact of digital transformation based on the degree of technological innovation of companies. According to the Classification of Strategic Emerging Industries (2018) and the definition of high-tech industry by the Organization for Economic Cooperation and Development (OECD), this paper determines the industry codes of high-tech listed companies against the Guidelines for Industry Classification of Listed

**Table 6. Heterogeneity tests.**

| Explanatory Variables | Explained variables | | | | | |
|---|---|---|---|---|---|---|
| | **(1)** | **(2)** | **(3)** | **(4)** | **(5)** | **(6)** |
| $DT$ | 0.1274 (0.1354) | 0.4152*** (0.1441) | 0.1274 (0.1354) | 0.6605*** (0.2178) | 0.5165** (0.2275) | 0.2582 (0.1575) |
| Constant term | -5.9766*** (0.3166) | -6.2436*** (0.2601) | -5.9766*** (0.3166) | -6.3000*** (0.3044) | -6.1019*** (0.2451) | -6.7708*** (0.3027) |
| Control variables | YES | YES | YES | YES | YES | YES |
| Individual fixed effects | YES | YES | YES | YES | YES | YES |
| Year fixed effects | YES | YES | YES | YES | YES | YES |
| $N$ | 8706 | 18399 | 13296 | 13808 | 18318 | 8722 |
| $R^2$ | 0.7576 | 0.7309 | 0.7576 | 0.7352 | 0.7646 | 0.7535 |
| **Explanatory Variables** | **Explained variables** | | | | | |
| | **(7)** | **(8)** | **(9)** | **(10)** | | |
| $DT$ | 13.2701*** (3.7970) | 0.4813*** (0.1767) | -0.5194** (0.2079) | 0.0609 (0.0402) | | |
| Constant term | -4.0132 (4.0627) | -4.2268*** (0.7620) | -0.8240 (0.9098) | -3.2071*** (0.6770) | | |
| Control variables | YES | YES | YES | YES | | |
| Individual fixed effects | YES | YES | YES | YES | | |
| Year fixed effects | YES | YES | YES | YES | | |
| $N$ | 236 | 1723 | 1875 | 1364 | | |
| $R^2$ | 0.5331 | 0.6591 | 0.6410 | 0.6512 | | |

Companies (2012 Revision). We group regressions for high-tech type companies and non-high-tech type companies, and the regression results are presented in columns (3) and (4) of Table 6. Among them, column (3) is the regression result for high-tech companies, and the estimated coefficient of digital transformation is 0.1274, which does not pass the significance test at 10% level. Column (4) is the regression result for non-high-tech companies with an estimated coefficient of 0.6605 and it passes the significance test at the 5% level. Possible reasons for this result are: it is more challenging for high-tech companies to update their production technology and equipment, so the digital transformation measures fail to significantly affect productivity in the short term; while non-high-tech companies restructure their business operation model by implementing digital transformation to accelerate the rate of internal information flow and enhance resource utilization efficiency, thus significantly improving productivity levels.

**4.4.2 Industry heterogeneity test.** In this paper, we perform group regressions based on the degree of competition in the industries. We use the Herfindahl Index (HHI) to measure the degree of industry competition. The higher the value of the HHI, the lower the degree of industry competition and the higher the degree of industry concentration. In this paper, the mean value of Herfindahl index for all industries in each year is used as the group classification criterion, and industries with Herfindahl index values below the mean are classified as the group with high industry competition, while industries with values above the mean are classified as the group with low industry competition. The specific empirical regression results are shown in columns (5) and (6) of Table 6, where column (5) shows the regression results for the subsample with a high degree of industry competition and column (6) shows the regression results for the subsample with a low degree of industry competition. The estimated coefficient of digital transformation is significantly positive at the 5% level with a coefficient of 0.5165 for the group with high industry competition; while in the group with low industry competition, the estimated coefficient for digital transformation is 0.2582, which does not pass the 10% significance level test. The regression results presented in these two columns indicate that the higher the industry competition, the more significant is the effect of digital transformation on total factor productivity improvement. Possible reasons for this result may be that, compared with companies in industries with low industry competition, companies in industries with high industry competition are more likely to be influenced by malicious defaults by upstream and downstream companies in the supply chain, negatively affecting their production and operation, whereas the introduction of advanced digital technology reduces default risk and helps companies to complete delivery or terminate contracts in time, thus improving the efficiency of production and operation.

In addition, in order to test the heterogeneous effects of digital transformation on different industries, we conducted regression analyses for the coal mining industry, pharmaceutical manufacturing industry, chemical raw materials and chemical products manufacturing industry, and software and information technology services industry, respectively. The results are shown in columns (7)-(10) of Table 6, with column (7) displaying the regression results for the coal mining industry, where digital transformation significantly improves the total factor productivity of listed firms in the coal mining industry. Column (8) shows the regression results for the pharmaceutical manufacturing industry, where digital transformation significantly improves the total factor productivity of listed firms in the pharmaceutical manufacturing industry. Column (9) shows the regression results for the chemical raw materials and chemical products manufacturing industry, where digital transformation significantly reduces the total factor productivity of listed firms in the chemical raw materials and chemical products manufacturing industry. Column (10) shows the regression results for the information technology services industry, where digital transformation does not significantly increase the total

factor productivity of listed firms in the information technology services industry. From the above results, it can be found that there is significant heterogeneity in the impact of digital transformation on different industries. The coal mining industry, as a representative type of mining industry, has a low level of digitization, and a certain degree of digital transformation can significantly improve productivity. Pharmaceutical manufacturing and chemical raw materials and chemical products manufacturing are both typical representative industries in the manufacturing industry. Pharmaceutical manufacturing has improved its productivity through digital transformation, while the implementation of digital transformation in the chemical raw materials and chemical products manufacturing industry has inhibited productivity. This may be due to the fact that the pharmaceutical manufacturing industry is technology-intensive and has a high demand for scientific and technological research and development. Digital transformation can effectively enhance the pharmaceutical manufacturing enterprise R & D efficiency and thus enhance the total factor productivity, while the chemical raw materials and chemical products manufacturing industry focuses on the manufacture of basic chemical raw materials and fertilizer manufacturing, vulnerable to macroeconomic fluctuations and macroeconomic policies, the enterprise elimination rate is high, the enterprise invests a lot of resources in digital transformation will crowd out the resources of the production sector so as to negatively affect the traditional production and management activities of the enterprise. activities. The information technology service industry, as a highly digitized industry in the service sector, has basically completed the digitization of its production and business processes, and therefore, the continued implementation of digital transformation cannot play a significant positive contributing role.

## 5. Mechanism analysis

### 5.1 Technological innovation role mechanism

The theoretical analysis discussed in this paper earlier indicates that digital transformation can promote companies' technological innovation output and innovation efficiency and improves total factor productivity. In order to verify the role of technological innovation in the process of digital transformation affecting total factor productivity, we use companies' corporate R&D input, technological innovation output and innovation efficiency as the explained variables, while other variables are consistent with the benchmark regression Eq (1). For the mechanism testing part we also use the 2SLS model for the regression, and the instrumental variable used is the regional mean of the firm's digital transformation. The regression results are presented in columns (1)-(3) of Table 7. In column (1), the explained variable is the level of corporate R&D investment, measured by dividing corporate current R&D investment by total assets, and the coefficient of digital transformation is significantly positive at the 5% level, indicating that

**Table 7. Mechanism tests of technological innovation and operating costs.**

| Explanatory Variables | Explained Variables | | | | |
|---|---|---|---|---|---|
| | (1) | (2) | (3) | (4) | (5) |
| $DT$ | 0.0068** (0.0034) | 0.1798*** (0.0578) | 0.0637** (0.0255) | -0.0913*** (0.0339) | -0.0949*** (0.0187) |
| Constant term | 0.0095 (0.0066) | -12.1957*** (0.5246) | -0.4367*** (0.0281) | 0.8935*** (0.0563) | 0.2700*** (0.0221) |
| Control variables | YES | YES | YES | YES | YES |
| Individual fixed effects | YES | YES | YES | YES | YES |
| Year fixed effects | YES | YES | YES | YES | YES |
| $N$ | 26982 | 26982 | 23518 | 27188 | 27188 |
| $R^2$ | 0.3591 | 0.3947 | 0.2711 | 0.3382 | 0.2777 |

digital transformation initiatives can significantly improve corporate R&D dynamics. In column (2), the explained variable is companies' technological innovation output, and the estimated coefficient of digital transformation is significantly positive as measured by the natural logarithm of the number of companies' current patent applications plus one. In column (3), the explained variable is companies' innovation efficiency, which is calculated by dividing the total number of invention, utility model and design patent applications plus the natural logarithm of 1 by the natural logarithm of R&D expenditures plus 1. The estimated coefficient of digital transformation remains significantly positive. The above empirical results show that the implementation of digital transformation initiatives by companies significantly improves companies' technological innovation capabilities in terms of innovation inputs, innovation outputs, and innovation efficiency, which further improves companies' total factor productivity.

## 5.2 Operating cost role mechanism

Columns (4) and (5) of Table 7 reflect the results of testing the impact mechanism of operating costs. We analyze the impact of digital transformation on companies' operating costs in terms of both the overhead rate and the operating cost rate. The explained variable in column (4) is the operating cost rate, which is calculated by dividing the main operating cost by the main operating revenue, and the estimated coefficient of digital transformation is negative at the 5% significance level. The explained variable in column (5) is the administrative expense ratio, calculated by dividing the administrative expenses by the main operating income. The estimated coefficient of digital transformation is significantly negative and the result indicates that the implementation of digital transformation strategy by firms helps them to increase their productivity level mainly by reducing the main operating costs and overheads.

## 5.3 Resource allocation efficiency mechanism

This paper verifies that digital transformation improves total factor productivity by improving company resource allocation efficiency in terms of both labor investment efficiency and supply chain operation efficiency. Firstly, as for labor investment efficiency, the theoretical analysis of this paper indicates that digital transformation measures will improve the labor investment efficiency of companies, thus improving productivity. To test this mechanism of action, we use company labor investment efficiency as the explained variable, and the other explanatory variables are consistent with the benchmark regression model (1). In particular, the company labor investment efficiency variable is constructed as follows: referring to Jung et al. (2014) [38], the percentage change in the number of employees is treated as the company's labor investment, the rate of change in the number of employees is used as the explained variable, and the explanatory variables are a series of company-level economic variables, which are put into an annual-industry fixed effects to regress. The absolute value of the residuals obtained from the estimated model is the efficiency of labor investment, a focus of this paper. The regression results are presented in column (1) of Table 8, and the estimated coefficient of digital transformation is significantly negative, which indicates that digital transformation can improve the labor investment efficiency of companies. The reasons are the deep application of digital technology facilitates the exchange and aggregation of information among various departments within the company; it also simplifies corporate information transmission channels and enhance information transparency. Increased information transparency can help management level make optimal employment decisions based on information, and it is easier for corporate regulators to monitor management's hiring practices to reduce over- or under-employment.

In this paper, we then examine the resource allocation efficiency mechanism from the perspective of operational efficiency from supply chain. We choose the inventory turnover ratio

**Table 8. Mechanism tests of resource allocation efficiency and human capital structure.**

| Explanatory Variables | Explained Variables | | | |
|---|---|---|---|---|
| | (1) | (2) | (3) | (4) |
| DT | -0.0521* (0.0306) | 0.5559 (0.5728) | 0.4816*** (0.1422) | 1.4275*** (0.4517) |
| Constant term | 0.2610*** (0.0623) | -5.7040*** (0.3740) | 0.2275*** (0.0606) | -30.5170*** (5.3268) |
| Control variables | YES | YES | YES | YES |
| Individual fixed effects | YES | YES | YES | YES |
| Year fixed effects | YES | YES | YES | YES |
| N | 19915 | 6541 | 20647 | 24828 |
| $R^2$ | 0.0602 | 0.7255 | 0.7792 | 0.4109 |

to represent the supply chain operational efficiency of an company, which is calculated by dividing the operating cost by the average of the opening inventory balance and the closing inventory balance of an company. The theoretical analysis of this paper indicates that digital transformation can help companies improve the efficiency of resource utilization from supply chain and thus improve the total factor productivity level. We choose to use group regression to test the mechanism of the benchmark regression model (1), and the specific regression results are shown in columns (2) and (3) of Table 8. Where column (2) is the regression result for the group with strong supply chain operation capability, the estimated coefficient for digital transformation is positive but does not pass the 10% significance test. Column (3) shows the regression results for the group with low supply chain operational capabilities, and the estimated coefficient on digital transformation is significantly positive at the 1% level. From the results presented in these two columns, it is clear that when supply chain operational efficiency is low, the effect of digital transformation on the total factor productivity of companies is more significant, which is in line with our theoretical expectation. These tests suggest that digital transformation can improve the total factor productivity level by enhancing the efficiency of companies' supply chain operations.

## 5.4 Human capital structure mechanism

To test the mechanism of the effect of digital transformation in improving the human capital structure of companies and thus affecting total factor productivity, this paper uses the human capital structure of companies as the explained variable, and the other variables are the same as in the benchmark regression model (1) equation. To reflect the educational level of the labor force, we measure the human capital structure of companies by the proportion of employees with bachelor's degree or above. The higher the value of this variable, the higher the degree of specialization and skill knowledge reserve in the labor factor, and the easier it is for companies to leap to the higher end of the value chain. The specific regression results are shown in column (4) of Table 8. The estimated coefficient of digital transformation is significantly positive at the 1% level, indicating that the digital transformation initiatives adopted by companies significantly raise the educational level of their workforce and optimize the human capital structure. The use of digital technologies and digital platforms reduces the demand for low-end labor, promoting companies to employ more professionals with digital technology backgrounds that will help them improve their production technologies and services, as well as their productivity. Meanwhile, as companies continue to add high-end human resources, the various skills and expertise available within the company can further promote collaborative innovation in different departments and thus increase total factor productivity.

## 6. Conclusion and countermeasure suggestions

### 6.1 Conclusion

The improvement in the total factor productivity of companies is a key concern for academics and government policy makers. As the world enters the digital era where countries hope to realize new economic growth points through digital transformation, it is crucial to study whether and how digital transformation can affect the total factor productivity of microeconomic agents companies.

This paper explains the intrinsic mechanism of digital transformation affecting the total factor productivity of companies at the theoretical level, and then empirically examines the impact of digital transformation on the total factor productivity of companies and its channels of action using panel data of listed companies in China. The empirical study of this paper draws the following conclusions: first, digital transformation significantly increases the total factor productivity level of companies, and the robustness test verifies the reliability of this conclusion; second, the heterogeneity test in this paper finds that the effect of digital transformation on total factor productivity is relatively larger for private companies, non-high-tech companies, and companies in highly competitive industries; third, digital transformation improves total factor productivity by promoting company technological innovation, reducing operational costs, improving resource allocation efficiency, and improving the human capital structure of companies to enhance total factor productivity.

### 6.2 Countermeasure suggestions

The findings of this paper imply that to bring out the positive effects of digital transformation on total factor productivity, companies and governments need to take differentiated initiatives based on company characteristics to release the development dividends from digital transformation.

First, governments should pay attention to the positive impact of digital transformation on total factor productivity. In the era of digital economy, a government plays a leading role in supporting the construction of digital infrastructure, creating a platform for digital technology innovation and application, and accelerating the deployment of digital strategies. For example, national key laboratory for digital technology led by governments can promote the research and development of new digital infrastructure and promote the application of emerging digital technologies to other industries and regions. Besides, digital transformation can reduce business operating costs and help companies to alleviate the rising cost crisis. Governments should encourage private companies to implement digital transformation, help private companies to introduce, develop and apply emerging digital technologies, and set up special financial subsidies for private companies and companies in some industries with fierce competition to help achieving digital and intelligent transformation. Meanwhile, governments should implement relevant policies to help companies realize the role of digital transformation, guide them to combine their own resource endowment and industry characteristics to find a suited digital development path, and encourage them to consider their own financial reality to develop digital transformation investment budget plans.

Second, companies should accelerate the application of digital technology innovation. The empirical results of this paper show that the behavior of Chinese listed companies implementing digital transformation can promote corporate R&D investment, innovation output and innovation efficiency, which in turn promote their total factor productivity growth. As digital transformation is gaining popularity, companies should take the initiative to strengthen their R&D investment and external cooperation in digital technology, accelerate the in-depth

application of digital technology in scenarios such as procurement, product development, production and sales, warehousing and logistics, customer service, finance and personnel management, promote the synergistic development of application innovation and technological innovation, and sustain the positive effect of digital technology on company transformation and upgrading. At the same time, companies should formulate special talent recruitment plans based on their own digital transformation needs, facilitate the introduction of digital technology professionals, optimize the human resource structure and accelerate knowledge overflow.

## Author Contributions

**Writing – original draft:** Zhonghao Lei.

**Writing – review & editing:** Dongmei Wang.

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
