## [Decision Letter · Decision Letter 0]

7 Jul 2023

PONE-D-23-17641Digital Transformation and Total Factor Productivity: Evidence from ChinaPLOS ONE

Dear Dr. Wang,

Thank you for submitting your manuscript to PLOS ONE. After careful consideration, we feel that it has merit but does not fully meet PLOS ONE’s publication criteria as it currently stands. Therefore, we invite you to submit a revised version of the manuscript that addresses the points raised during the review process.

We look forward to receiving your revised manuscript.

Kind regards,

Tao Bu

Academic Editor

PLOS ONE

Journal Requirements:

   "We acknowledge the 2023 Gansu Provincial Department of Education College Teachers’ Innovation Fund Project (No. 2023A-281)"

Additional Editor Comments:

I now have two referee report and have also read the paper myself. The referee thinks the paper has addressed some important questions, and the referee also unanimous in his views on the suitability of the paper for the PLOS ONE and recommend a major revision. I find myself in agreement with his views and so have decided to accept for publication after carrying out these “major revision”.

We appreciate you submitting your manuscript to PLOS ONE and thank you for giving us the opportunity to consider your work.  

Kind regards,    

Tao Bu  

Academic Editor  

Reviewer #1:

The authors have presented a well written article regarding digital transformation of companies. I would like to request if they have the variation in the extent of transformation based on the different sectors than just a high tech and low tech for eg: does health sector companies lack the transformation compared to telecom sector.

Reviewer #2:

This paper examines the effects of digital transformation in improving total factor productivity based on data from A-shared companies in China. The authors find that digital transformation significantly improves total factor productivity. Such effects are stronger for private companies, non-tech companies, and companies with a high degree of industry competition. Mechanism analysis is also performed. Overall, I think it is an interesting paper.

My major concern is the validity of their results. In my opinion, what the authors identified is simply the relationship between digital transformation and total factor productivity, not the causal effect.

As also mentioned by the authors, one threat to their estimation is the omitted variable bias. Thus, they used two IVs to address the problem. However, I do not think the IVs are good enough.

Specifically, the authors use the mean of digital values at the city and industry level as IVs but fail to explain the reasons why they do so. In other words, the authors need to explicitly verify whether these two IVs satisfy the “exclusion restriction” assumption (intuitively AND mathematically).

Moreover, see Table 3 and Table 4, baseline results and IVs are completely different in terms of magnitude, meaning that there does exist omitted variable bias. Thus, IVs should be considered as the primary model. I do not see why the authors continue to use the baseline model in their later analysis.

Other minor concerns:

1. Full name of the abbreviation should be given at its first appearance. See “OP and LP”.

2. The IV approach at least should be mentioned in the introduction part.

Reviewers' comments:

Reviewer's Responses to Questions

**Comments to the Author**

1. Is the manuscript technically sound, and do the data support the conclusions?

Reviewer #1: Yes

Reviewer #2: Partly

2. Has the statistical analysis been performed appropriately and rigorously? 

Reviewer #1: Yes

Reviewer #2: No

3. Have the authors made all data underlying the findings in their manuscript fully available?

Reviewer #1: Yes

Reviewer #2: Yes

4. Is the manuscript presented in an intelligible fashion and written in standard English?

Reviewer #1: Yes

Reviewer #2: Yes

5. Review Comments to the Author

Reviewer #1: The authors have presented a well written article regarding digital transformation of companies. I would like to request if they have the variation in the extent of transformation based on the different sectors than just a high tech and low tech for eg: does health sector companies lack the transformation compared to telecom sector.

Reviewer #2: This paper examines the effects of digital transformation in improving total factor productivity based on data from A-shared companies in China. The authors find that digital transformation significantly improves total factor productivity. Such effects are stronger for private companies, non-tech companies, and companies with a high degree of industry competition. Mechanism analysis is also performed. Overall, I think it is an interesting paper.

My major concern is the validity of their results. In my opinion, what the authors identified is simply the relationship between digital transformation and total factor productivity, not the causal effect.

As also mentioned by the authors, one threat to their estimation is the omitted variable bias. Thus, they used two IVs to address the problem. However, I do not think the IVs are good enough.

Specifically, the authors use the mean of digital values at the city and industry level as IVs but fail to explain the reasons why they do so. In other words, the authors need to explicitly verify whether these two IVs satisfy the “exclusion restriction” assumption (intuitively AND mathematically).

Moreover, see Table 3 and Table 4, baseline results and IVs are completely different in terms of magnitude, meaning that there does exist omitted variable bias. Thus, IVs should be considered as the primary model. I do not see why the authors continue to use the baseline model in their later analysis.

Other minor concerns:

1. Full name of the abbreviation should be given at its first appearance. See “OP and LP”.

2. The IV approach at least should be mentioned in the introduction part.

6. PLOS authors have the option to publish the peer review history of their article (what does this mean?). If published, this will include your full peer review and any attached files.

Reviewer #1: **Yes: **Sudha Chandrashekar

Reviewer #2: No

---

## [Author Response · Author response to Decision Letter 0]

26 Jul 2023

We would like to express our heartfelt gratitude to three anonymous reviewers for their valuable feedback. Our manuscript, titled “Digital Transformation and Total Factor Productivity: Empirical Evidence from China”, benefited significantly from the constructive comments and insights from the review team. Based on the suggestions received, we have made careful revisions to the original manuscript. In the revised manuscript, all changes are marked in red. In addition, we have also carefully proofread this manuscript for typographical, grammatical, and other errors. We hope the revised manuscript is able to meet your standard of quality and address the concerns raised by the reviewers. In the following, you will find our detailed point-by-point responses to the reviewers’ comments.

Reviewer #1

Major Comments

The authors have presented a well written article regarding digital transformation of companies. I would like to request if they have the variation in the extent of transformation based on the different sectors than just a high tech and low tech for eg: does health sector companies lack the transformation compared to telecom sector.

Response：

Thank you for focusing on whether there are differences in the extent of digital transformation across industries, which is an important issue that we have overlooked. In order to investigate the heterogeneity across industries, we conducted regressions on coal mining, pharmaceutical manufacturing, chemical raw materials and chemical products manufacturing, and software and information technology services in the revised manuscript, and found that there is significant heterogeneity in the impact of digital transformation on total factor productivity across industries. The specific alteration is as follows:

…

Line 443

In addition, in order to test the heterogeneous effects of digital transformation on different industries, we conducted regression analyses for the coal mining industry, pharmaceutical manufacturing industry, chemical raw materials and chemical products manufacturing industry, and software and information technology services industry, respectively. The results are shown in columns (7)-(10) of Table 6, with column (7) displaying the regression results for the coal mining industry, where digital transformation significantly improves the total factor productivity of listed firms in the coal mining industry. Column (8) shows the regression results for the pharmaceutical manufacturing industry, where digital transformation significantly improves the total factor productivity of listed firms in the pharmaceutical manufacturing industry. Column (9) shows the regression results for the chemical raw materials and chemical products manufacturing industry, where digital transformation significantly reduces the total factor productivity of listed firms in the chemical raw materials and chemical products manufacturing industry. Column (10) shows the regression results for the information technology services industry, where digital transformation does not significantly increase the total factor productivity of listed firms in the information technology services industry. From the above results, it can be found that there is significant heterogeneity in the impact of digital transformation on different industries. The coal mining industry, as a representative type of mining industry, has a low level of digitization, and a certain degree of digital transformation can significantly improve productivity. Pharmaceutical manufacturing and chemical raw materials and chemical products manufacturing are both typical representative industries in the manufacturing industry. Pharmaceutical manufacturing has improved its productivity through digital transformation, while the implementation of digital transformation in the chemical raw materials and chemical products manufacturing industry has inhibited productivity. This may be due to the fact that the pharmaceutical manufacturing industry is technology-intensive and has a high demand for scientific and technological research and development. Digital transformation can effectively enhance the pharmaceutical manufacturing enterprise R & D efficiency and thus enhance the total factor productivity, while the chemical raw materials and chemical products manufacturing industry focuses on the manufacture of basic chemical raw materials and fertilizer manufacturing, vulnerable to macroeconomic fluctuations and macroeconomic policies, the enterprise elimination rate is high, the enterprise invests a lot of resources in digital transformation will crowd out the resources of the production sector so as to negatively affect the traditional production and management activities of the enterprise. activities. The information technology service industry, as a highly digitized industry in the service sector, has basically completed the digitization of its production and business processes, and therefore, the continued implementation of digital transformation cannot play a significant positive contributing role.

Table 6

Heterogeneity tests

Explanatory

Variables Explained variables

 （1） （2） （3） （4） （5） （6）

DT 0.1274

（0.1354） 0.4152***

（0.1441） 0.1274

（0.1354） 0.6605***

（0.2178） 0.5165**

（0.2275） 0.2582

（0.1575）

Constant term -5.9766***

（0.3166） -6.2436***

（0.2601） -5.9766***

（0.3166） -6.3000***

（0.3044） -6.1019***

（0.2451） -6.7708***

（0.3027）

Control variables YES YES YES YES YES YES

Individual fixed effects YES YES YES YES YES YES

Year fixed effects YES YES YES YES YES YES

N 8706 18399 13296 13808 18318 8722

R^2 0.7576 0.7309 0.7576 0.7352 0.7646 0.7535

Explanatory

Variables Explained variables

 （7） （8） （9） （10） 

DT 13.2701***

（3.7970） 0.4813***

（0.1767） -0.5194**

（0.2079） 0.0609

（0.0402） 

Constant term -4.0132

（4.0627） -4.2268***

（0.7620） -0.8240

（0.9098） -3.2071***

（0.6770） 

Control variables YES YES YES YES 

Individual fixed effects YES YES YES YES 

Year fixed effects YES YES YES YES 

N 236 1723 1875 1364 

R^2 0.5331 0.6591 0.6410 0.6512 

…

Reviewer #2

1: My major concern is the validity of their results. In my opinion, what the authors identified is simply the relationship between digital transformation and total factor productivity, not the causal effect. 

Response：

Thank you for your insightful feedback, I understand your concern very well. Regarding this issue, we adopted the instrumental variable of digital transformation to study the causal effect between digital transformation and total factor productivity. The modified section is as follows:

…

Line83

…

while we construct instrumental variables and apply two-stage least squares to deal with possible endogeneity issues..

…

3: As also mentioned by the authors, one threat to their estimation is the omitted variable bias. Thus, they used two IVs to address the problem. However, I do not think the IVs are good enough.

Response：

We appreciate your vigilance. We choose the mean of digital values at the city and industry level as IVs for two reasons: first, the degree of digital transformation of individual firms is affected by the digital transformation of the region and industry in which the firms are located, so the instrumental variables we choose satisfy the correlation assumption. Second, the regional mean and industry mean of firms' digital transformation have been homogenized and are weakly correlated with firms' total factor productivity. In this sense, the instrumental variables we selected are not related to the current production and business activities of the firms and satisfy the exclusivity assumption. In addition, we checked the plausibility of instrumental variables, all of which were statistically significant. This is elaborated in our revised manuscript. The specific alteration is as follows:

…

Line 278

We choose the mean of digital values at the city and industry level as IVs for two reasons: first, the degree of digital transformation of individual firms is affected by the digital transformation of the region and industry in which the firms are located, so the instrumental variables we choose satisfy the correlation assumption. Second, the regional mean and industry mean of firms' digital transformation have been homogenized and are weakly correlated with firms' total factor productivity. In this sense, the instrumental variables we selected are not related to the current production and business activities of the firms and satisfy the exclusivity assumption.

…

4: Specifically, the authors use the mean of digital values at the city and industry level as IVs but fail to explain the reasons why they do so. In other words, the authors need to explicitly verify whether these two IVs satisfy the “exclusion restriction” assumption (intuitively AND mathematically).

Thank you for drawing attention to this. We sincerely apologize for not explaining the reasons for using the city and industry level means of digital values as IVs, the main reasons why we use them as instrumental variables can be found in the previous Response. In addition, we examine the plausibility of the instrumental variables. The first is to test the correlation hypothesis, the first stage regression results estimated by 2SLS show that the coefficients of the regional mean and industry mean of enterprise digital transformation are 0.9792 and 0.6283, respectively, which both pass the significance test at the 1% level, indicating that the instrumental variables are strongly correlated with the core explanatory variables. We then test whether the exclusivity assumption is valid. In general, the exclusionary hypothesis is difficult to test directly, and the presence of a weak instrumental variable problem will invalidate the exclusionary hypothesis. Therefore, we focus on the 2SLS estimated correlation statistic to determine whether there is a weak instrumental variable problem. This is elaborated in our revised manuscript. The specific alteration is as follows:

…

Line 291

We next tested the plausibility of the instrumental variables. The first is to test the correlation hypothesis, the first stage regression results estimated by 2SLS show that the coefficients of the regional mean and industry mean of enterprise digital transformation are 0.9792 and 0.6283, respectively, which both pass the significance test at the 1% level, indicating that the instrumental variables are strongly correlated with the core explanatory variables. We then test whether the exclusivity assumption is valid. In general, the exclusionary hypothesis is difficult to test directly, and the presence of a weak instrumental variable problem will invalidate the exclusionary hypothesis. Therefore, we focus on the 2SLS estimated correlation statistic to determine whether there is a weak instrumental variable problem.

…

5: Moreover, see Table 3 and Table 4, baseline results and IVs are completely different in terms of magnitude, meaning that there does exist omitted variable bias. Thus, IVs should be considered as the primary model. I do not see why the authors continue to use the baseline model in their later analysis.

Response：

Thank you for this fantastic and useful suggestion. To mitigate endogeneity, we used instrumental variables with reference to common practice in economics and used instrumental variables to deal with possible endogeneity in heterogeneity tests and mechanism tests. The specific alteration is as follows: 

…

Line 510

The explained variable in column (5) is the administrative expense ratio, calculated by dividing the administrative expenses by the main operating income. The estimated coefficient of digital transformation is significantly negative and the result indicates that the implementation of digital transformation strategy by firms helps them to increase their productivity level mainly by reducing the main operating costs and overheads.

Table 7

Mechanism tests of technological innovation and operating costs

Explanatory

Variables Explained Variables

 （1） （2） （3） （4） （5）

DT 0.0068**

（0.0034） 0.1798***

（0.0578） 0.0637**

（0.0255） -0.0913***

（0.0339） -0.0949***

（0.0187）

Constant term 0.0095

（0.0066） -12.1957***

（0.5246） -0.4367***

（0.0281） 0.8935***

（0.0563） 0.2700***

（0.0221）

Control variables YES YES YES YES YES

Individual fixed effects YES YES YES YES YES

Year fixed effects YES YES YES YES YES

N 26982 26982 23518 27188 27188

R^2 0.3591 0.3947 0.2711 0.3382 0.2777

 Line 547

Where column (2) is the regression result for the group with strong supply chain operation capability, the estimated coefficient for digital transformation is positive but does not pass the 10% significance test. Column (3) shows the regression results for the group with low supply chain operational capabilities, and the estimated coefficient on digital transformation is significantly positive at the 1% level.

Table 8

Mechanism tests of resource allocation efficiency and human capital structure

Explanatory

Variables Explained Variables

 （1） （2） （3） （4）

DT -0.0521*

（0.0306） 0.5559

（0.5728） 0.4816***

（0.1422） 1.4275***

（0.4517）

Constant term 0.2610***

（0.0623） -5.7040***

（0.3740） 0.2275***

（0.0606） -30.5170***

（5.3268）

Control variables YES YES YES YES

Individual fixed effects YES YES YES YES

Year fixed effects YES YES YES YES

N 19915 6541 20647 24828

R^2 0.0602 0.7255 0.7792 0.4109

…

Minor Comments:

1: Full name of the abbreviation should be given at its first appearance. See “OP and LP”.

We sincerely apologize for not giving the full names of the abbreviations at its first appearance. "OP" refers to the semiparametric treatment proposed by Olley and Pakes, and "LP" refers to the semiparametric treatment proposed by Levinsohn and Petrin. We have clarified this in the revised manuscript. The specific alteration is as follows:

 …

Line 203

Explained variables (TFP): Empirical studies in the micro domain usually use the semiparametric treatment proposed by Olley and Pakes (1996) (referred to as the OP method) as well as the semiparametric treatment proposed by Levinsohn and Petrin (2003) (referred to as the LP method) to measure the total factor productivity of firms, 

…

2: The IV approach at least should be mentioned in the introduction part.

Response：

We sincerely apologize for not mentioning the IV approach in the introduction part. In the revised manuscript, we have refined this in the introduction part. The specific alteration is as follows:

…

Line 83

while we construct instrumental variables and apply two-stage least squares to deal with possible endogeneity issues

---

## [Decision Letter · Decision Letter 1]

2 Aug 2023

PONE-D-23-17641R1Digital Transformation and Total Factor Productivity: Evidence from ChinaPLOS ONE

Dear Dr. Wang,

Thank you for submitting your manuscript to PLOS ONE. After careful consideration, we feel that it has merit but does not fully meet PLOS ONE’s publication criteria as it currently stands. Therefore, we invite you to submit a revised version of the manuscript that addresses the points raised during the review process.

We look forward to receiving your revised manuscript.

Kind regards,

Tao Bu

Academic Editor

PLOS ONE

Journal Requirements:

Reviewers' comments:

Reviewer's Responses to Questions

**Comments to the Author**

1. If the authors have adequately addressed your comments raised in a previous round of review and you feel that this manuscript is now acceptable for publication, you may indicate that here to bypass the “Comments to the Author” section, enter your conflict of interest statement in the “Confidential to Editor” section, and submit your "Accept" recommendation.

Reviewer #2: (No Response)

2. Is the manuscript technically sound, and do the data support the conclusions?

Reviewer #2: (No Response)

3. Has the statistical analysis been performed appropriately and rigorously? 

Reviewer #2: (No Response)

4. Have the authors made all data underlying the findings in their manuscript fully available?

Reviewer #2: (No Response)

5. Is the manuscript presented in an intelligible fashion and written in standard English?

Reviewer #2: (No Response)

6. Review Comments to the Author

Reviewer #2: The authors did an excellent work revising the paper and I very much appreciate their efforts. Now the construction of IVs makes more sense than the original version.

In my initial comments, I was concerned that the baseline results (OLS results) are significantly different from those obtained from IV regressions, indicating the OLS approach may suffer from omitted variable bias (endogeneity issues). The authors adopted IVs for heterogeneity and mechanism analysis, which is fine. But I still recommend the authors briefly discuss the differences between OLS estimates and IVs estimates and make it clear that the endogeneity issue is indeed a threat to their estimation and thus they use IVs in later sections.

7. PLOS authors have the option to publish the peer review history of their article (what does this mean?). If published, this will include your full peer review and any attached files.

Reviewer #2: No

---

## [Author Response · Author response to Decision Letter 1]

14 Aug 2023

Thank you for your valuable comments, which help us a lot. The differences between OLS and IVs estimation are briefly discussed in the paper, in addition to the fact that we have made it clear in the paper that the endogeneity problem does threaten our estimation results, so we use IVs in the later sections. The specific alteration is as follows:

…

Line 304

It should be emphasized that the IVs estimates obtained in this paper through instrumental variables are local average treatment effects that exclude the part of exogenous changes included in the explanatory variable digital transformation. That is, changes in firms' digital transformation due to bidirectional causation are excluded, and thus the regression coefficients are significantly different from the OLS coefficient values. In addition, because the endogeneity issue does affect the study estimates, we use IVs estimation in both the heterogeneity test and the mechanism test later.

…

---

## [Decision Letter · Decision Letter 2]

3 Oct 2023

Digital Transformation and Total Factor Productivity: Evidence from China

PONE-D-23-17641R2

Dear Dr. Wang,

We’re pleased to inform you that your manuscript has been judged scientifically suitable for publication and will be formally accepted for publication once it meets all outstanding technical requirements.

Kind regards,

Donato Morea

Academic Editor

PLOS ONE

Reviewers' comments:

Reviewer's Responses to Questions

**Comments to the Author**

1. If the authors have adequately addressed your comments raised in a previous round of review and you feel that this manuscript is now acceptable for publication, you may indicate that here to bypass the “Comments to the Author” section, enter your conflict of interest statement in the “Confidential to Editor” section, and submit your "Accept" recommendation.

Reviewer #2: (No Response)

2. Is the manuscript technically sound, and do the data support the conclusions?

Reviewer #2: (No Response)

3. Has the statistical analysis been performed appropriately and rigorously? 

Reviewer #2: (No Response)

4. Have the authors made all data underlying the findings in their manuscript fully available?

Reviewer #2: (No Response)

5. Is the manuscript presented in an intelligible fashion and written in standard English?

Reviewer #2: (No Response)

6. Review Comments to the Author

Reviewer #2: (No Response)

7. PLOS authors have the option to publish the peer review history of their article (what does this mean?). If published, this will include your full peer review and any attached files.

Reviewer #2: No

---

## [Editor Report · Acceptance letter]

8 Oct 2023

PONE-D-23-17641R2 

Digital Transformation and Total Factor Productivity: Empirical evidence from China 

Dear Dr. Wang:

I'm pleased to inform you that your manuscript has been deemed suitable for publication in PLOS ONE. Congratulations! Your manuscript is now with our production department. 

Kind regards, 

on behalf of

Professor (Assistant) Donato Morea 

Academic Editor

PLOS ONE